# Association of leukocyte composition ratios from blood methylation with cancer mortality outcomes
Ziwen Fan [1], Dominic Edelmann[2], Zitong Zhao[1], Bruno Christian Köhler [3,4], Michael Hoffmeister [1] & Hermann Brenner [1,5,6,7] ✉

## Abstract

**Background** Leukocyte composition ratios derived from blood genome-wide methylation (DNAm-derived LCRs), reflecting systemic inflammation, remain unclear in relation to various mortality outcomes.

**Methods** We performed an epigenome-wide analysis to identify the association of DNAm-derived LCRs with all-cause mortality, cancer-specific mortality, and lung-cancer-specific mortality in a large prospective cohort study with 17 years follow-up.

**Results** Strong associations of multiple LCRs are seen for all mortality outcomes. The neutrophil-to-B-cell ratio was strongly associated with all-cause mortality (HR per SD increase, 1.20; 95% CI, 1.10–1.31), the neutrophil-to-lymphocyte ratio with cancer-specific mortality (HR, 1.28; 1.11–1.49), and the lymphocyte-to-monocyte ratio with lung-cancer-specific mortality (HR, 0.53; 0.38–0.75). The consistency of HR estimations across 11-year, 14-year, and 17-year follow-ups reinforces these findings. Several LCRs show stronger associations in females and younger participants.

**Conclusions** Our study identifies DNAm-derived LCRs as particularly useful measures for quantifying cancer mortality risk over long-term follow-ups exceeding a decade.

## Plain Language Summary

This study looked at how certain blood cell patterns estimated from DNA analysis, called leukocyte composition ratios (LCRs), relate to the risk of death from any cause, cancer, and lung cancer. We used data from a long-term study in Germany that followed people aged 50 to 75 for 17 years. LCRs were measured using a new method in blood samples. The results showed that some LCRs are strongly linked to higher or lower risk of death, especially from cancer and lung cancer. These findings suggest that assessment of LCRs could help identify people at higher risk of cancer death. Understanding these blood markers may improve health monitoring and disease prevention efforts in the future.

Inflammation is a fundamental biological hallmark of ageing, characterized by systemic, sterile, low-grade chronic inflammation in the elderly[1]. Chronic inflammation is critically involved in all stages of cancer development, supporting cancer initiation, promoting cancer progression, and supporting metastatic diffusion[2]. Leukocyte composition ratios (LCRs) may reflect this systemic inflammation, which tends to increase with age[3]. Extensive research has illustrated that elevated levels of inflammatory status, indicated by increased neutrophil-to-lymphocyte ratio (NLR) and decreased lymphocyte-to-monocyte ratio (LMR), associated with elevated mortality rates among cancer patients[4–6]. However, evidence on the association between LCRs and long-term cancer-specific mortality within the general population keeps being limited[7,8].

Genome-wide DNA methylation (DNAm) analysis of whole blood samples provides an innovative method for the accurate and detailed quantification of LCRs[9–11], which were previously determined by direct cell counts (cytological LCRs) subject to imprecision and variability[12]. The leukocyte composition metrics derived from methylation data are routinely employed as covariates in epigenome-wide association studies to account for variations in cell types among individuals. Emerging evidence underscores the prognostic significance of these metrics, as calculated LCRs have been associated with survival outcomes in various cancers. For example, increased DNAm-derived NLR has been associated with poorer survival in patients with glioma[13], breast cancer[14], lung cancer[15], and pediatric medulloblastoma[16].

[1]Division of Clinical Epidemiology and Aging Research, German Cancer Research Center (DKFZ), Heidelberg, Germany. [2]Division of Biostatistics, German Cancer Research Center (DKFZ), Heidelberg, Germany. [3]Liver Cancer Center Heidelberg, Heidelberg University Hospital, Heidelberg, Germany. [4]Department of Medical Oncology, National Center for Tumor Diseases, Heidelberg University Hospital, Heidelberg, Germany. [5]NCT Heidelberg, National Center for Tumor Diseases (NCT), a partnership between DKFZ and University Hospital, Heidelberg, Germany. [6]Division of Preventive Oncology, German Cancer Research Center (DKFZ), Heidelberg, Germany. [7]German Cancer Consortium (DKTK), German Cancer Research Center (DKFZ), Heidelberg, Germany. ✉e-mail: h.brenner@Dkfz-Heidelberg.de

This study aims to investigate the associations between DNAm-derived LCRs and mortality outcomes, including all-cause mortality, cancer-specific mortality and lung-cancer-specific mortality within a cohort of older adults from a general population. Additionally, we investigate how the length of follow-up affects the prognostic value of these DNAm-derived LCRs by analyzing the data over different follow-up periods.

## Methods

### Study design and population

Our analysis is based on data from the ongoing ESTHER study (Epidemiologische Studie zu Chancen der Verhütung, Früherkennung und optimierten Therapie chronischer Erkrankungen in der älteren Bevölkerung, DRKS00014028), a population-based cohort study with baseline and long-term follow-up data collected in Saarland, Germany. Comprehensive details of this cohort, including study design and participants demographics, have been previously reported[17–19]. Briefly, the ESTHER study involved the recruitment of 9940 participants aged 50–75 years at baseline, conducted between July 2000 and December 2002. Enrollment occurred through general practitioners during routine health checkups, with follow-ups scheduled every two or three years. Participants provide information on sociodemographic factors, health status, lifestyle habits, and major disease history through standardized self-administered questionnaires. General practitioners contribute detailed medical records, including disease diagnoses and medication prescriptions. Peripheral blood samples are taken at baseline and each follow-up and preserved at −80 °C for future analysis. Ethical approval was granted by the ethics committees of Heidelberg University's medical faculty and the medical board of Saarland, with all participants providing written informed consent.

DNA methylation analyses by epigenome-wide methylation arrays were performed in baseline blood samples of a total of 4195 participants in four batches in the context of preliminary projects between Oct 2012 and July 2019[17,20–22]: **Subset I** comprised 1034 participants randomly selected from the overall study. **Subset II** was selected based on a case-cohort design within 7244 participants and included three groups: 260 participants diagnosed with breast, lung, or colorectal cancer during 14 years of follow-up; 542 participants randomly chosen from 1329 who died during the same period; and 739 participants selected as a sub-cohort without regard to cancer or mortality status. **Subset III** consisted of 864 participants from a case-cohort design within 2499 individuals, including 406 who died by March 2013 and 548 randomly chosen as a sub-cohort irrespective of death status. **Subset IV** included the first 500 women and 500 men consecutively enrolled during the first six months. Each subset (I, II, III, and IV) included 1034, 1297, 864, and 1000 participants respectively. These subsets were independent and non-overlapping, ensuring diverse and robust sampling for our methylation analysis.

### DNA methylation assessment

The DNAm profiles for subsets I and II were analyzed using the Infinium Methylation EPIC BeadChip kit (EPIC, Illumina Inc., San Diego, CA, USA), while those for subsets III and IV were assessed with the Infinium Human Methylation450K BeadChip Assay (450 K, Illumina Inc., San Diego, CA, USA). Both assays were performed according to the manufacturer's instructions and underwent preprocessing (Fig. S1)[23]. Samples failing to meet quality control standards were excluded. Exclusion criteria included failed control matrix, more than $10^5$ undetected CpGs (detection $p$ values greater than 0.01), mismatched sex, contamination by single-nucleotide polymorphism (SNP) outliers, and outliers in principal component analysis. Dye correction was applied to normalize beta values, which range from 0 (completely unmethylated) to 1 (completely methylated). Finally, a total of 3886 participants passed the quality criteria were included in the analysis.

### Leukocyte composition estimation

We estimated leukocyte composition using three algorithms: Houseman's algorithm (HOU) with the *minfi* R package's "estimateCellCounts" function[9], and the Salas' (SAL) and LOLIPOP (LOL) algorithms through the

*ewastools* R package's "estimateLC" function[10,11]. These three deconvolution algorithms were selected because they are publicly available and specifically developed for use with whole blood DNA methylation data. Specifically, we only applied SAL to subsets I and II, as it is designed for EPIC DNAm data. The HOU and SAL algorithms estimate the percentages of several cell types, including granulocytes (GR), monocytes (MO), natural killer (NK) cells, B cells (B), CD4 + T cells (CD4T), and CD8 + T cells (CD8T). The LOL algorithm focuses on estimating neutrophils (NE), eosinophils (EO), basophils (BA), monocytes (MO), and lymphocytes (LY). Each algorithm ensures that the total leukocyte composition sums to one. To handle any zero or negative values in cell proportions, we replaced them with a small value ($10^{-10}$). For LY proportions estimated by the HOU and SAL algorithms, we summed the NK, B, CD4T, and CD8T proportions.

### Calculation of DNAm-derived LCRs

We used DNAm-derived leukocyte composition data to calculate four inflammation-related biomarkers exclusively involving leukocytes (LCRs): neutrophil/lymphocyte ratio (NLR), lymphocyte/monocyte ratio (LMR), (sum of neutrophils and monocytes)/lymphocytes ratio (sNMLR), and (product of neutrophils and monocytes)/lymphocytes ratio (pNMLR)[24,25]. Additionally, we computed ratios for specific lymphocyte subtypes, such as the ratio of neutrophils to B cells (NBR). For calculations involving the HOU and SAL algorithms, NE values were substituted with GR value. A total of 44 LCRs were determined from leukocyte compositions estimated by three deconvolution algorithms, enabling the calculation of previously validated inflammation-related ratios (e.g., NLR, LMR, sNMLR, pNMLR) associated with cancer outcomes. We trimmed the leukocyte composition values by setting the outliers to the upper bound (third quartile + 1.5 × interquartile range [IQR]) or lower bound (first quartile − 1.5 × IQR), outliers were replaced with the corresponding upper or lower bound.

### Statistical analysis

We used descriptive statistics to summarize the baseline characteristics of the participants. Categorical covariates were described as absolute and relative frequencies, and continuous variables were described using medians and IQRs.

In subsets I and IV, multivariable Cox regression analyses were conducted to explore the associations between LCRs and 17-year follow-up mortalities, including all-cause mortality, cancer-specific mortality, and lung-cancer-specific mortality. Mortality from other specific cancers was not individually assessed due to case number limitations. For subsets II and III, which employed a case-cohort design, weighted Cox regression models were used, accounting for the sampling design with inverse probability weights[26,27]. The proportional hazards assumption was checked by scaled Schoenfeld residuals plots. Survival time was defined from baseline to death or the end of follow-up. Analyses were adjusted for age, sex, batch effects, smoking status (never, former, current smoker), alcohol consumption (grams/day), body mass index (kg/m²), educational level (≤9 years, 10–11 years, ≥12 years), physical activity (inactive, low, medium, high), history of cardiovascular diseases, diabetes, and hypertension. Observations with missing values for any of the variables were excluded from the analyses. Dose-response relationships were examined based on LCR quartiles.

Given the variations in DNAm profiles assessment, owing to different time periods and DNAm chips used across the four subsets, results were presented separately and combined through random-effects meta-analysis. The meta-analysis for the SAL algorithm, which derives LCRs exclusively from EPIC DNAm data, was limited to subsets I and II.

Sensitivity analyses were performed restricting the follow-up periods to 11- and 14-years, respectively. Additional models for lung cancer controlled for smoking pack-years instead of smoking status. We conduct additional sensitivity analyses by further adjusting—separately—for renal failure, asthma, chronic obstructive pulmonary disease (COPD), rheumatoid arthritis, neurodermatitis, and NSAID use in the multivariable Cox

**Table 1 | Baseline characteristics of the study population**

| Characteristics | Overall (N = 3886) | Subset I (N = 935) | Subset II (N = 1117) | Subset III (N = 857) | Subset IV (N = 977) |
|---|---|---|---|---|---|
| Age (years, median [IQR]) | 63 [57.0, 68.0] | 62.0 [57.0, 67.0] | 63.0 [58.0, 68.0] | 63.0 [58.0, 68.0] | 62.0 [57.0, 67.0] |
| Gender (female; N, %) | 2116 (54.5) | 520 (55.6) | 631 (56.5) | 471 (55.0) | 494 (50.6) |
| Smoke (N, %)[a] | | | | | |
| Never | 1829 (48.3) | 477 (52.1) | 516 (47.6) | 376 (45.3) | 460 (48.2) |
| Former | 1227 (32.4) | 289 (31.6) | 333 (30.7) | 286 (34.5) | 319 (33.4) |
| Current | 728 (19.2) | 149 (16.3) | 235 (21.7) | 168 (20.2) | 176 (18.4) |
| Alcohol consumption (g/d, median [IQR]) [b] | 5.1 [0.0, 13.5] | 5.1 [0.0, 14.2] | 3.7 [0.0, 13.1] | 5.8 [0.0, 12.8] | 5.8 [0.0, 14.3] |
| BMI (kg/m², median [IQR]) | 27.3 [24.8, 30.1] | 27.2 [24.8, 30.1] | 27.4 [25.0, 30.4] | 27.0 [24.4, 30.1] | 27.4 [24.9, 30.2] |
| Physical activity (N, %) [c] | | | | | |
| Inactive | 874 (22.6) | 198 (21.2) | 284 (25.6) | 193 (22.6) | 199 (20.4) |
| Sedentary | 1797 (46.4) | 447 (47.9) | 501 (45.1) | 423 (49.5) | 426 (43.6) |
| Vigorously active | 1204 (31.1) | 289 (30.9) | 326 (29.3) | 238 (27.9) | 351 (36.0) |
| Education (N, %) [d] | | | | | |
| ≤9 y | 2879 (76.2) | 697 (76.4) | 841 (77.9) | 639 (76.4) | 702 (73.7) |
| 10–11 y | 525 (13.9) | 115 (12.6) | 141 (13.1) | 120 (14.4) | 149 (15.6) |
| ≥12 y | 376 (9.9) | 100 (11.0) | 97 (9.0) | 77 (9.2) | 102 (10.7) |
| Prevalent CVD cases (N, %) | 889 (22.9) | 191 (20.4) | 262 (23.5) | 220 (25.7) | 216 (22.1) |
| Prevalent diabetes cases (N, %) [e] | 686 (17.7) | 130 (13.9) | 212 (19.0) | 181 (21.1) | 163 (16.7) |
| Prevalent hypertension cases (N, %) | 2334 (60.1) | 529 (56.6) | 692 (62.0) | 528 (61.6) | 585 (59.9) |
| Median follow-up time (years) | 17.9 (17.9–18) | 17.4 (17.3–17.5) | 17.5 (17.4–17.6) | 18 (18–18) | 18.3 (18.3–18.4) |
| Deaths (N) | 1643 | 244 | 573 | 486 | 340 |
| Cancer deaths (N) | 583 | 69 | 209 | 191 | 114 |
| Lung cancer deaths (N) | 151 | 18 | 59 | 47 | 27 |

[a]Data missing for 20, 33, 27, 22, and 102 participants in subset I, II, III, IV, and across all study populations in ESTHER.
[b]Data missing for 95, 130, 82, 77, and 384 in subset I, II, III, IV, and across all study populations in ESTHER.
[c]Data missing for 1, 6, 3, 1, and 11 in subset I, II, III, IV, and across all study populations in ESTHER.
[d]Data missing for 23, 38, 21, 24, and 106 in subset I, II, III, IV, and across all study populations in ESTHER.
[e]Data missing for 2, 1, and 3 in subset II, III, and across all study populations in ESTHER.
*IQR* interquartile range, *BMI* body mass index, *CVD* cardiovascular disease.

regression models examining associations between DNAm-derived LCRs and 17-year all-cause, cancer-specific, and lung cancer–specific mortality (see Supplementary Data 1–3).

Subgroup analyses were conducted to evaluate the association of DNAm-derived LCRs with mortality by sex and age groups (younger, 50–64 years old; older, 65–75 years old). We also tested for interactions of DNAm-derived LCRs with sex and age groups.

All statistical analyses were executed using the R programming language (version 4.1.2) and R Studio (version 1.4.1717; Boston, USA). The *ewastools* R package was used for DNAm preprocessing and leukocyte composition estimation[28], the *minfi* package for additional leukocyte composition estimation[29], and the *survival* package for Cox regression analysis[30]. Statistical tests were two-tailed with a significance level of 0.05.

## Results
### Characteristics of study population
Table 1 present the baseline characteristics of the ESTHER study population across different subsets. The median (IQR) age of participants was 63 (57–68) years; 2116 were females (54.5%). In all for four subsets, the median age was between 62 and 63 years, the majority of participants in each subset was female. Approximately half of the participants in each subset were either former or current smokers. Median daily alcohol consumption in the four subsets was between 3.7 g and 5.8 g, the median BMI was between 27.0 and 27.4 kg/m². Over the 17 years of follow-up, a total of 1643 deaths were recorded, of which 583 and 151 were due to any cancer and lung cancer. The number of deaths recorded in each subset were as follows: 244 in subset I, 573 in subset II, 486 in subset III, and 340 in subset IV.

### Comparison of prognostic value of DNAm-derived LCRs
Figure 1 illustrates the associations between DNAm-derived LCRs and three mortality outcomes: all-cause mortality, cancer-specific mortality, and lung-cancer-specific mortality. Higher DNAm-derived NLR, sNMLR, and pNMLR were associated with an increased risk of mortality outcomes, while a higher DNAm-derived LMR was associated with a decreased risk of mortality outcomes. This pattern was consistent when these markers were constructed with lymphocyte subtypes. While the point estimates for the hazard ratios (HR) per standard deviation (SD) increase for all-cause mortality and cancer-specific mortality were similar, the associations with cancer-specific mortality displayed wider 95% confidence intervals (CI) compared to those with all-cause mortality. The associations with lung-cancer-specific mortality were stronger. The strongest associations for all-cause mortality, cancer-specific mortality, and lung-cancer-specific mortality were observed with NBR, NLR, and LMR derived from Salas' algorithm, yielding multivariate-adjusted HRs (per SD increase) of 1.20 (95% CI, 1.10–1.31), 1.28 (1.11–1.49), and 0.53 (0.38–0.75), respectively.

Dose-response relationships for LCRs with mortality outcomes are detailed in Figs. S2–6. Most LCRs demonstrated monotonic dose-response relationships with all mortality types. Particularly strong dose-relationships were noted for various LCRs with lung-cancer-specific mortality, especially for LMR derived from Salas' algorithm, which showed a markedly lower lung cancer mortality for the highest quartile compared to the lowest (HR 0.16, 95% CI 0.06–0.40).

### Subgroup analyses and sensitivity analyses
Across 44 DNAm-derived LCRs, most showed similar associations with the three mortality outcomes for men and women, and younger and older

**Fig. 1 | Associations of DNAm-derived LCRs with mortality outcomes across all subsets.** This figure illustrates the associations of LCRs derived by three different algorithms (colors) with mortality outcomes in all subsets. **a** All-cause mortality, **b** cancer-specific mortality, **c** lung-cancer–specific mortality. The models were adjusted for age, sex, batch, smoking status, alcohol consumption, body mass index, educational level, physical activity, history of cardiovascular diseases, diabetes, and hypertension. Meta-analysis was performed for subsets I and II using the SAL algorithm and for all subsets using the HOU and LOL algorithms. Dot and error bars depict hazard ratios (HR) along with their 95% confidence intervals (CI). Abbreviations: LCR leukocyte composition ratio, HR hazard ratio, CI confidence interval, SD standard deviation, NLR neutrophil-to-lymphocyte ratio, sNMLR ratio of sum of neutrophil and monocytes divided by lymphocyte, pNMLR ratio of product of neutrophil and monocytes divided by lymphocyte, LMR lymphocyte-to-monocyte ratio, HOU Housman's algorithm, SAL Salas' algorithm, LOL LOLIPOP algorithm, B B cell, CD4T CD4+ T cell, CD8T CD8+ T cell, NK natural killer cell.

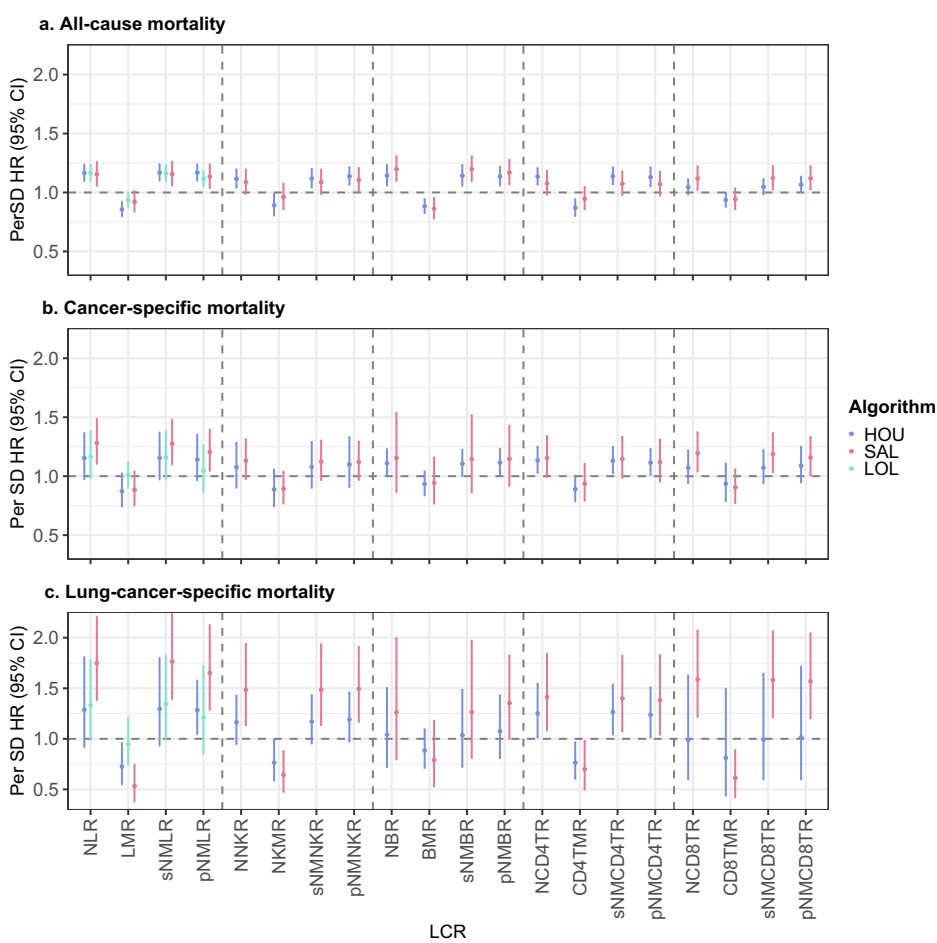

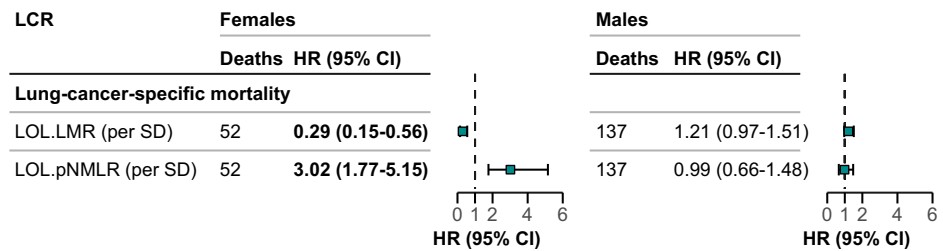

**Fig. 2 | Sex-Specific subgroup analysis of associations of DNAm-Derived LCRs with mortality outcomes across all subsets.** This figure displays a sex-specific subgroup analysis, focusing only on DNAm-derived LCRs that have demonstrated statistically significant interactions (P-trend < 0.05) with sex across all subsets. The models were adjusted for age, batch, smoking status, alcohol consumption, body mass index, educational level, physical activity, history of cardiovascular diseases, diabetes, and hypertension. Meta-analysis was conducted for all subsets using the LOL algorithm. Values shown in bold indicate statistically significant results. Green squares and error bars depict hazard ratios (HR) along with their 95% confidence intervals (CI). Abbreviations: LCR leukocyte composition ratio, HR hazard ratio, CI confidence interval, SD standard deviation, pNMLR ratio of product of neutrophil and monocytes divided by lymphocyte, LMR lymphocyte-to-monocyte ratio, LOL LOLIPOP algorithm.

participants (Figs. S7, 8). Nevertheless, tests for interactions for some DNAm-derived LCRs with sex and age reached statistical significance (P-trend < 0.05). The DNAm-derived LMR and pNMLR estimated by LOL algorithms showed stronger associations with lung-cancer specific mortality in females than in males (Fig. 2). Twenty-four DNAm-derived LCRs showed significant interactions by age (Fig. 3), and 20 of them showed stronger associations with the three mortality outcomes in younger participants than in older ones. Four LCRs (pNMBR, NCD4TR, sNMCD4TR, and pNMCD4TR) estimated by Salas' algorithm showed particularly strong associations with lung cancer mortality in younger participants, with HRs around 2 for per standard deviation increase in these LCRs.

We conducted sensitivity analyses by adjusting smoking pack-years instead of smoking status in the association of LCRs with LCM, showing highly consistent associations (Fig. S9). Further sensitivity analyses for the association of DNAm-derived LCRs with 11-year (Fig. S10) and 14-year follow-up mortalities (Fig. S11) showed consistent HR estimations with those obtained with 17-year follow-up for all DNAm-derived LCRs.

## Discussion

In this prospective cohort study involving 3886 older adults across a 17-year follow-up, we found that various DNAm-derived LCRs were strongly and significantly associated with all-cause mortality, cancer-specific mortality,

**Fig. 3 | Age-Specific subgroup analysis of associations of DNAm-Derived LCRs with mortality outcomes across all subsets.** This figure displays an age-specific subgroup analysis (younger: 50–64 years old, older: 65–75 years old), focusing only on DNAm-derived LCRs that have demonstrated statistically significant interactions (P-trend < 0.05) with age across all subsets. The models were adjusted for sex, batch, smoking status, alcohol consumption, body mass index, educational level, physical activity, history of cardiovascular diseases, diabetes, and hypertension. Meta-analysis of subset I, II for SAL algorithms, Meta-analyses were conducted for subsets I and II using the SAL algorithm and for all subsets using the HOU and LOL algorithms. Values shown in bold indicate statistically significant results. Green squares and error bars depict hazard ratios (HR) along with their 95% confidence intervals (CI). Abbreviations: LCR leukocyte composition ratio, HR hazard ratio, CI confidence interval, SD standard deviation, HOU Housman's algorithm, SAL Salas' algorithm, LOL LOLIPOP algorithm, NLR neutrophil-to-lymphocyte ratio, sNMLR ratio of sum of neutrophil and monocytes divided by lymphocyte, pNMLR ratio of product of neutrophil and monocytes divided by lymphocyte, LMR lymphocyte-to-monocyte ratio, B B cell, CD4T CD4$^+$ T cell, NK natural killer cell.

| LCR | Younger participants | | Older participants | |
|---|---|---|---|---|
| | Deaths | HR (95% CI) | Deaths | HR (95% CI) |
| **All-cause mortality** | | | | |
| SAL.NBR (per SD) | 351 | **1.33 (1.16-1.52)** | 493 | **1.17 (1.03-1.31)** |
| SAL.sNMBR (per SD) | 351 | **1.33 (1.16-1.52)** | 493 | **1.17 (1.03-1.32)** |
| SAL.NCD4TR (per SD) | 351 | **1.29 (1.12-1.48)** | 493 | 0.96 (0.85-1.10) |
| SAL.CD4TMR (per SD) | 351 | **0.70 (0.57-0.87)** | 493 | 1.11 (0.90-1.37) |
| SAL.sNMCD4TR (per SD) | 351 | **1.30 (1.13-1.49)** | 493 | 0.96 (0.84-1.09) |
| SAL.pNMCD4TR (per SD) | 351 | **1.34 (1.17-1.54)** | 493 | 0.95 (0.83-1.08) |
| HOU.NNKR (per SD) | 697 | 0.99 (0.86-1.14) | 946 | **1.18 (1.08-1.28)** |
| HOU.sNMNKR (per SD) | 697 | 1.00 (0.86-1.15) | 946 | **1.18 (1.08-1.28)** |
| **Cancer-specific mortality** | | | | |
| HOU.NBR (per SD) | 332 | **1.30 (1.13-1.49)** | 322 | 0.99 (0.86-1.16) |
| HOU.sNMBR (per SD) | 332 | **1.29 (1.13-1.48)** | 322 | 0.99 (0.85-1.15) |
| HOU.pNMBR (per SD) | 332 | **1.32 (1.15-1.50)** | 322 | 0.99 (0.86-1.16) |
| HOU.NCD4TR (per SD) | 332 | **1.27 (1.11-1.45)** | 322 | 1.07 (0.92-1.25) |
| HOU.sNMCD4TR (per SD) | 332 | **1.27 (1.11-1.46)** | 322 | 1.07 (0.91-1.24) |
| HOU.pNMCD4TR (per SD) | 332 | **1.27 (1.06-1.51)** | 322 | 1.02 (0.88-1.19) |
| LOL.NLR (per SD) | 332 | **1.27 (1.03-1.58)** | 322 | 1.11 (0.94-1.32) |
| LOL.sNMLR (per SD) | 332 | **1.27 (1.01-1.60)** | 322 | 1.10 (0.92-1.31) |
| **Lung-cancer-specific mortality** | | | | |
| SAL.pNMBR (per SD) | 73 | **1.99 (1.36-2.90)** | 41 | 0.87 (0.41-1.86) |
| SAL.NCD4TR (per SD) | 73 | **2.00 (1.44-2.78)** | 41 | 1.15 (0.66-2.01) |
| SAL.sNMCD4TR (per SD) | 73 | **1.99 (1.43-2.77)** | 41 | 1.10 (0.64-1.92) |
| SAL.pNMCD4TR (per SD) | 73 | **2.01 (1.41-2.86)** | 41 | 0.94 (0.58-1.52) |
| HOU.NBR (per SD) | 115 | 1.34 (0.86-2.07) | 74 | 0.64 (0.33-1.26) |
| HOU.sNMBR (per SD) | 115 | 1.32 (0.87-2.00) | 74 | 0.64 (0.34-1.22) |
| HOU.pNMBR (per SD) | 115 | **1.44 (1.01-2.04)** | 74 | 0.69 (0.44-1.08) |
| HOU.pNMCD4TR (per SD) | 115 | **1.61 (1.23-2.10)** | 74 | 1.07 (0.72-1.59) |

0 1 2 3 HR (95% CI)       0 1 2 3 HR (95% CI)

and lung-cancer-specific mortality. Specifically, higher DNAm-derived NLR, sNMLR, pNMLR and lower LMR, as well as LCRs constructed with lymphocyte subtypes, were associated with poor prognoses. In general, these associations were more pronounced with lung-cancer-specific mortality than with all-cause mortality and all-cancer mortality. The consistency of HR estimations across 11-year, 14-year and 17-year follow-up periods further reinforces the robustness of these findings. Additionally, two DNAm-derived LCRs specifically predictive of lung-cancer-specific mortality showed stronger associations in females than in males. Twenty DNAm-derived LCRs associated with different mortality outcomes demonstrated stronger associations in younger participants compared to older ones. Particularly strong associations were seen for four LCRs (pNMBR, NCD4TR, sNMCD4TR, and pNMCD4TR) estimated by Salas' algorithm with lung cancer mortality among younger participants.

Cytological NLR has been shown to be significantly associated with all-cause mortality in general populations in the Netherlands[7] and the United States[8]. This association likely reflects the influence of pre-existing conditions that can lead to early mortality, as well as the role of inflammation or impaired immune function in the progression of long-term diseases. In contrast, the association between cytological NLR and long-term cancer-specific survival has not been significant, which could be expected since its prognostic value may primarily depend on the cancer's presence and factors specific to the local tumor environment, such as tumor-infiltrating neutrophils and lymphocytes[31–35]. These factors are typically absent in individuals without cancer. Previous systematic reviews and meta-analyses have highlighted that both higher cytological NLR and lower cytological LMR, with a cut-off value ranging from two to five, are associated with increased overall mortality in patients with various solid tumors[4,5,36–38].

Our findings are consistent with previous results showing associations between LCRs and all-cause mortality. However, our results also show

significant associations of DNAm-derived LCRs with cancer-specific mortality, particularly lung-cancer-specific mortality, which contrasts with the typically weak associations observed with cytological LCRs. DNAm-derived LCRs appear to offer advantages in evaluating long-term cancer-specific mortality due to their relations to both cancer incidence and methylation profile changes related to cancer mortality. Several studies examining both cytological LCRs in general populations[39] and DNAm-derived LCRs among high-risk individuals for lung cancer[40] have reported significant associations with cancer incidence. DNAm-derived LCRs are potentially associated to changes in methylation profiles, which are associated with both all-cause and cancer-specific mortality[17,22]. Notably, strong associations between cytological NLR and lung cancer–specific mortality have also been observed, even in individuals without a prior lung cancer diagnosis[41]. In clinical settings, elevated NLR, LMR, and other inflammation-related markers are widely used as prognostic indicators for lung cancer, particularly among patients with stage IV non-small cell lung cancer[42]. From a mechanistic perspective, chronic exposure to airborne pathogens or toxic agents can trigger overproduction of reactive oxygen/nitrogen species (ROS/RNS), leading to sustained inflammation and lung tissue injury[43]. Consistent with this, chronic inflammatory states—such as smoking, chronic obstructive pulmonary disease (COPD), and bronchitis—are well-established risk factors for lung cancer and may contribute to both aberrant methylation and cancer progression through shared molecular mechanisms[44–46].

DNAm-derived LCRs offer a distinct advantage by enabling the analysis of LCRs involving specific lymphocyte subtypes, a capability not present in cytological LCR analyses. Recent research has shown a marked association of CD4T and B cells, as determined by the SAL algorithm, with both all-cause mortality and CRC-specific mortality among CRC patients[47]. Similarly, in our own research, we observed that the ratio of neutrophils to B

cells (NBR) exhibited the strongest prognostic value for evaluating all-cause mortality.

Cytological NLR is positively correlated with aging, and typically, females exhibit lower NLR values than males, suggesting reduced inflammation levels in younger individuals and females[3]. In line with this, our findings indicate that several DNAm-derived LCRs had more pronounced associations with mortality outcomes in females compared to males, and in younger individuals compared to older ones. This observation could suggest that a lower baseline level of inflammation enhances the sensitivity of DNAm-derived LCRs in detecting aggressive disease patterns and predicting poorer outcomes. Moreover, we found that the relationships between DNAm-derived LCRs and mortality outcomes remained stable across follow-ups of 11, 14, and 17 years. Consequently, monitoring DNAm-derived LCRs from an early age could not only reflect aging-related systemic inflammation but also act as reliable predictive markers over extended follow-up periods exceeding a decade.

The major strengths of this study include its population-based cohort study design, long-term mortality follow-up, comprehensive collection of health data, meta-analyzing data according to methylation measurement batch. Nonetheless, our study has several limitations. Firstly, the most crucial DNAm-derived LCRs are estimated using Salas' algorithms, which were only applicable to subsets I and II that utilized the DNAm EPIC array. This limitation might reduce the statistical power and generalizability of our results due to a smaller sample size. Secondly, the absence of cytologic LCR measurements prevents direct comparison between cytologic and DNAm-derived LCRs in mortality outcome evaluation. Moreover, DNAm microarrays are a costlier method of measuring LCR compared to direct measurement from absolute blood counts, though these costs could potentially decrease with the advancement and application of targeted DNAm marker technologies. Thirdly, the limited number of specific cancer death events other than lung cancer hindered analyses of DNAm-derived LCR's associations with other cancer mortality outcomes. Lastly, our study's reliance on a single DNAm assessment point means that repeated measurements over time in future could provide more comprehensive insights.

In conclusion, our study identified various DNAm-derived LCRs which are strongly associated with mortality outcomes. The DNAm-derived LCRs could be particularly useful measures to quantify all-cancer- and lung-cancer-specific mortality. Future research should focus on evaluating the utility of DNAm-derived LCRs in longitudinal aging studies that include younger participants and repeat DNAm measurements, as well as in clinical practice, in order to further explore prognostic value and potential clinical application in diverse settings. For example, the very strong associations of some of the LCRs with lung cancer mortality, especially at ages 50–64, which were observed even after adjustment for smoking in our study, may suggest their potential use for selecting high-risk people for lung cancer screening which should be explored in further research.

## Data availability
The datasets generated and/or analyzed during the current study are not publicly available due ethical and legal restrictions but are available from the corresponding author on reasonable request.

## Code availability
R codes for statistical analysis are available upon request.

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

## Acknowledgements

Dr. Fan was supported by a grant from the DKFZ clinician scientist program. The ESTHER study related work was supported by the Baden-Württemberg state Ministry of Science, Research and Arts (Stuttgart, Germany), the Federal Ministry of Education and Research (Berlin, Germany), the Federal Ministry of Family Affairs, Senior Citizens, Women and Youth (Berlin, Germany), the Saarland State Ministry of Health, Social Affairs, Women and the Family (Saarbrücken, Germany). The funding organizations had no role in the design and conduct of the study; collection, management, analysis, and interpretation of the data; preparation, review, or approval of the manuscript; and decision to submit the manuscript for publication. The authors thank the study participants and their general practitioners as well as laboratory and administrative staff of the ESTHER study team.

## Author contributions

Z.F. and H.B. had full access to all the data in the study and take responsibility for the integrity of the data and the accuracy of the data analysis. Concept and design: H.B. and Z.F. Acquisition, analysis, or interpretation of data: Z.F., D.E., Z.Z., B.C.K., M.H., and H.B. Drafting of the manuscript: Z.F. Critical revision of the manuscript for important intellectual content: Z.F., D.E., Z.Z., B.C.K., M.H., and H.B. Statistical analysis: Z.F. and D.E. Obtained funding: H.B. and Z.F. Administrative, technical, or material support: H.B. Supervision: H.B.

## Funding

## Competing interests

The authors declare no competing interests.
