## [Transparent Peer Review file · Communications Medicine]

Association of Leukocyte Composition Ratios from Blood Methylation with Cancer Mortality Outcomes

Corresponding Author: Dr Ziwen Fan

Version 0:

Reviewer comments:

Reviewer #1

(Remarks to the Author)

The authors present an analysis of DNA-based cell typing results from methylation deconvolution to test associations with mortality in a large prospective cohort using multiple subsets of participants. This large study makes important contributions to assessing the potential utility of DNA-based cell typing with DNA methylation measures and takes advantage of participants with long-term follow up data in the ESTHER cohort. Data analyses are rigorous and include appropriate consideration of potential confounding in multivariable models. Authors also include sensitivity analyses to restrict based on follow up time and also show analyses stratified by biological sex. The findings provide key information about capabilities of DNA methylation deconvolution that offer additional leukocyte subtypes compared with traditional clinical cell enumeration methods. I have no major concerns. It appears that the Sala's algorithm may actually be Salas' ?

Reviewer #2

(Remarks to the Author)

The manuscript investigates the prognostic value of DNAm-derived lymphocyte-to-cell ratios (LCRs) in predicting all-cause, cancer-specific, and lung cancer-specific mortality. The study is based on a large prospective cohort (ESTHER study, n=3,886) with 17 years of follow-up, providing longitudinal insights into the association between DNAm-derived inflammatory markers and mortality risks. While the study presents important findings, several points could be clarified or expanded to enhance its rigor and potential clinical relevance:

1. Although DNAm markers may precede disease progression, they could also be influenced by pre-existing health conditions. Could the authors clarify how they address this potential reverse causation issue? Have they considered additional health conditions that might contribute to altered DNAm patterns?
2. The authors have adjusted for smoking pack-years, but could other lifestyle and comorbidity factors also influence mortality risk? Additional sensitivity analyses considering chronic diseases (e.g., kidney disease, chronic inflammation, autoimmune conditions, or medication use) would help assess whether these factors modify the associations between DNAm-derived LCRs and mortality outcomes.
3. The findings suggest that DNAm-derived leukocyte composition ratios (LCRs) are strongly associated with all-cause, cancer-specific, and lung cancer-specific mortality. Given these significant associations. Could the authors discuss whether these DNAm-derived markers could serve as potential tools for clinical risk stratification? Are there thresholds or cutoffs for these LCRs that could be used for stratifying high-risk patients in a clinical setting?
4. In the methods and results, authors mentioned 44 DNAm-derived LCRs, I suppose that authors should have their reasons to select those 44 LCRs? Could authors explain the reason and add it to in the manuscript to make it more clear for the readers?
5. In the methods part (lines:134-136), "We trimmed the leukocyte composition values by setting the outliers to the upper bound", then how're the outliers?" Were they excluded or replaced?

Reviewer #3

(Remarks to the Author)

In this paper, Ziwen Fan et al. conducted calculation of Leukocyte Composition Ratios (LCR) from DNA methylation data of blood samples and evaluated associations between DNAm-derived LCR and cancer mortality. The authors found that

DNAm-derived LCRs were associated with all-cause mortality. Several kinds of ratios were also correlated with mortality. The data with long-term follow-up is very valuable. However, the reviewer thought that the authors at least need to show models to select high-risk patients for enriching clinical applications based on DNAm-derived LCRs because biological relevance was not enough in the findings of this paper.

Further, there are some points which should be clearly addressed.

Points;

1. The authors used three methods of DNAm-derived LCR estimation. Are they gold standard? If not, they should describe why they selected them.
2. In the Discussion section, the authors mentioned the absence of cytologic LCR measurements. However, they should validate DNAm-derived LCR compositions using different experimental methods and evaluate differences among them.
3. Why DNAm-derived LCRs more strongly associate with lung cancer mortality compared with others?
4. For Data availability, the authors could open the processed data (e.g. calculated LCRs).

Version 1:

Reviewer comments:

Reviewer #2

(Remarks to the Author)

The manuscript presents an insightful and rigorous investigation into the association between DNAm-derived leukocyte composition ratios (LCRs) and multiple mortality outcomes over a 17-year follow-up within a large prospective cohort. The use of epigenome-wide data to quantify systemic inflammation and its prognostic value is both novel and compelling. The major strengths include the long follow-up duration, comprehensive covariate adjustment, and stratified analyses across demographic subgroups.

The study's findings—especially the robust associations of NLR, LMR, and neutrophil-to-B-cell ratios with all-cause, cancer-specific, and lung cancer-specific mortality—are convincing and well-supported by sensitivity analyses. The consistency of hazard ratio estimates across multiple follow-up intervals enhances the credibility of the results.

The authors have addressed potential confounding and reverse causation appropriately. Their adjustments for chronic diseases and inflammation-related conditions, as well as the use of sensitivity analyses to account for comorbidities such as renal failure, asthma, and medication use, further reinforce the robustness of the findings. While some models were limited by missing data and low event counts, the consistency of results supports the main conclusions.

The decision to use quartile-based LCRs for dose-response analyses instead of predefined thresholds is justified and reflects a cautious approach given the variability in cutoffs across populations. This method enhances interpretability and facilitates future validation work.

Clarifications provided about the selection of 44 LCRs from three deconvolution algorithms and the trimming method for outlier handling have improved the transparency and reproducibility of the methods.

Overall, the study makes a valuable contribution to the field of epigenetic epidemiology and cancer prognostication. With minor revisions as incorporated, the manuscript is suitable for publication.

Reviewer #3

(Remarks to the Author)

The reviewer understood that it was challenging to establish a model. The authors' approach is more practical to capture the relationship between DNAm-derived LCRs and outcomes. They have addressed all of the concerns.

Point-by-point Response to the reviewers' comments of

Association of Leukocyte Composition Ratios from Blood Methylation with Cancer Mortality Outcomes: A Prospective Cohort Study

Reviewer #1

The authors present an analysis of DNA-based cell typing results from methylation deconvolution to test associations with mortality in a large prospective cohort using multiple subsets of participants. This large study makes important contributions to assessing the potential utility of DNA-based cell typing with DNA methylation measures and takes advantage of participants with long-term follow up data in the ESTHER cohort. Data analyses are rigorous and include appropriate consideration of potential confounding in multivariable models. Authors also include sensitivity analyses to restrict based on follow up time and also show analyses stratified by biological sex. The findings provide key information about capabilities of DNA methylation deconvolution that offer additional leukocyte subtypes compared with traditional clinical cell enumeration methods. I have no major concerns. It appears that the Sala's algorithm may actually be Salas' ?

Response

We are grateful to the reviewer for recognising the merits in our manuscript. We acknowledge the mistake use of Sala's and have corrected all the Sala's to Salas' accordingly.

Reviewer #2

The manuscript investigates the prognostic value of DNAm-derived lymphocyte-to-cell ratios (LCRs) in predicting all-cause, cancer-specific, and lung cancer-specific mortality. The study is based on a large prospective cohort (ESTHER study, n=3,886) with 17 years of follow-up, providing longitudinal insights into the association between DNAm-derived inflammatory markers and mortality risks. While the study presents important findings, several points could be clarified or expanded to enhance its rigor and potential clinical relevance:

Response

We express our appreciation to the reviewer for recognizing the strengths of our manuscript. Below, we present our point-to-point responses and revisions.

Point 1

1. Although DNAm markers may precede disease progression, they could also be influenced by pre-existing health conditions. Could the authors clarify how they address this potential reverse causation issue? Have they considered additional health conditions that might contribute to altered DNAm patterns?

Response

Thank you for this important comment. In our previous integrative analyses of clinical and epigenetic biomarkers of mortality—including cardiovascular and cancer-specific deaths—we found that age, sex, BMI, smoking, and alcohol consumption are well-established pre-existing health conditions that strongly predict cancer mortality (1). Furthermore, smoking emerged as one of the most influential factors associated with DNA methylation, particularly in the context of lung cancer (2-4). In the current analysis, we adjusted for comprehensive covariates including age, sex BMI, smoking and alcohol consumption in the Cox regression models. Given that DNAm-derived LCRs may reflect systemic inflammatory status, we further adjusted for chronic inflammation-related conditions such as diabetes, hypertension, and a history of cardiovascular events. In addition, blood samples were obtained at the time of the interview, and these collections were not conducted during acute infections. These adjustments help mitigate the potential influence of pre-existing health conditions and reduce the likelihood of reverse causation.

Point 2

2. The authors have adjusted for smoking pack-years, but could other lifestyle and comorbidity factors also influence mortality risk? Additional sensitivity analyses considering chronic diseases (e.g., kidney disease, chronic inflammation, autoimmune conditions, or medication use) would help assess whether these factors modify the associations between DNAm-derived LCRs and mortality outcomes.

Response

Thank you for this insightful comment. In our primary multivariable models, we adjusted for a range of sociodemographic and lifestyle-related factors, as well as chronic inflammation-related conditions, including diabetes, hypertension, and history of cardiovascular disease.

To address the reviewer’s suggestion more directly, we conducted additional sensitivity analyses by further adjusting—separately—for renal failure, asthma, chronic obstructive pulmonary disease (COPD), rheumatoid arthritis, neurodermatitis, and NSAID use in the multivariable Cox regression models examining associations between DNAm-derived LCRs and 17-year all-cause, cancer-specific, and lung cancer-specific mortality (see Response Tables 1–3).

However, due to a high rate of missing data for renal failure, COPD, and rheumatoid arthritis, these models are likely subject to considerable bias. Additionally, for some models, the small number of corresponding death events led to non-convergence of the Cox regression. For the sensitivity analyses additionally adjusted for asthma, neurodermatitis, and NSAID use, we observed similar results compared to the results from the comprehensively adjusted models. These findings support the robustness and stability of our main results.

Response Table 1. Characteristics of the study population for additional variates in sensitivity analysis

Characteristics	Overall (N = 3886)	Subset I (N=935)	Subset II (N=1117)	Subset III (N= 857)	Subset IV (N=977)
Prevalent renal failure (N, %) ^a	30 (2.1)	3 (0.7)	13 (3.2)	5 (2.1)	9 (2.3)
Prevalent asthma (N, %) ^b	233 (6.2)	48 (5.3)	70 (6.4)	40 (4.9)	75 (7.9)
Prevalent COPD (N, %) ^c	16 (1.1)	4 (0.9)	5 (1.3)	3 (1.2)	4 (1.0)
Prevalent rheumatoid arthritis (N, %) ^d	9 (0.7)	2 (0.5)	2 (0.6)	2 (1.0)	3 (0.9)
Prevalent neurodermatitis (N, %) ^e	142 (3.9)	33 (3.7)	46 (4.4)	30 (3.7)	33 (3.5)
Prevalent NSAIDs (N, %) ^f					
Prescribed Aspirin only	471 (12.1)	109 (11.7)	137 (12.3)	111 (13.0)	114 (11.7)
Prescribed other NSAID	137 (3.5)	24 (2.6)	43 (3.9)	27 (3.2)	43 (4.4)
Daily use of non- prescribed pain killer	204 (5.3)	47 (5.0)	69 (6.2)	42 (4.9)	46 (4.7)

^a Data missing for 513, 714, 624, 580, and 2431 participants in subset I, II, III, IV, and across all study populations in ESTHER.

^b Data missing for 23, 30, 38, 22, and 113 in subset I, II, III, IV, and across all study populations in ESTHER.

^c Data missing for 510, 719, 616, 586, and 2431 in subset I, II, III, IV, and across all study populations in ESTHER.

^d Data missing for 567, 761, 654, 633, and 2615 in subset I, II, III, IV, and across all study populations in ESTHER.

^e Data missing for 47, 73, 56, 43 and 219 in subset I, II, III, IV, and across all study populations in ESTHER.

^f Data missing for 1 and 1 in subset II and across all study populations in ESTHER.

Point 3

3. The findings suggest that DNAm-derived leukocyte composition ratios (LCRs) are strongly associated with all-cause, cancer-specific, and lung cancer-specific mortality. Given these significant associations. Could the authors discuss whether these DNAm-derived markers could serve as potential tools for clinical risk stratification? Are there thresholds or cutoffs for these LCRs that could be used for stratifying high-risk patients in a clinical setting?

Response

Given the known variability of LCR in optimal cutoffs across populations, even widely studied ratios such as NLR and LMR have shown a broad range of thresholds (e.g., between 2 and 5) (5). In addition, the ESTHER study includes four distinct subsets, each with differing baseline characteristics, which further complicates the definition of a universal cutoff. Therefore, we adopted a quartile-based approach to explore dose-response relationships between DNAm-derived LCRs and cancer outcomes. This strategy allows for consistent interpretation without relying on potentially unstable absolute thresholds. Future research will be essential to validate these markers for clinical risk stratification and to define standardized cutoffs.

Point 4

4. In the methods and results, authors mentioned 44 DNAm-derived LCRs, I suppose that authors should have their reasons to select those 44 LCRs? Could authors explain the reason and add it to in the manuscript to make it more clear for the readers?

Response

Thank you for this important question. The 44 DNAm-derived leukocyte composition ratios (LCRs) were selected based on three deconvolution algorithms that estimate proportions of granulocytes, monocytes, lymphocytes, and lymphocyte subtypes. These LCRs include key inflammation-related ratios such as neutrophil-to-lymphocyte ratio (NLR), lymphocyte-to-monocyte ratio (LMR), sum of neutrophils and monocytes to lymphocytes ratio (sNMLR), and product of neutrophils and monocytes to lymphocytes ratio (pNMLR), which have been previously associated with elevated mortality in cancer patients (6-9). This rationale has now been added to the Methods section (Page 5, Lines 19-21) for clarity.

Point 5

5. In the methods part (lines:134-136), “We trimmed the leukocyte composition values by setting the outliers to the upper bound”, then how’re the outliers? Were they excluded or replaced?

Response

The outliers were not excluded but replaced. Specifically, values exceeding the upper bound (defined as the third quartile + $1.5 \times$ interquartile range [IQR]) were set to the upper bound, and values below the lower bound (first quartile - $1.5 \times$ IQR) were set to the lower bound. We have now clarified this procedure in the Methods section (Page 5, Lines 23-24).

Reviewer #3

In this paper, Ziwen Fan et al. conducted calculation of Leukocyte Composition Ratios (LCR) from DNA methylation data of blood samples and evaluated associations between DNAm-derived LCR and cancer mortality. The authors found that DNAm-derived LCRs were associated with all-cause mortality. Several kinds of ratios were also correlated with mortality. The data with long-term follow-up is very valuable. However, the reviewer thought that the authors at least need to show models to select high-risk patients for enriching clinical applications based on DNAm-derived LCRs because biological relevance was not enough in the findings of this paper.

Response

Thank you for your thoughtful feedback on our study.

We acknowledge the importance of identifying high-risk patients for clinical applications. However, due to the known variability of optimal LCR cutoffs across populations and the heterogeneity among the four distinct subgroups in the ESTHER study, establishing a model with a universal threshold for risk stratification is currently challenging. Therefore, we employed a quartile-based approach to capture dose-response relationships between DNAm-derived LCRs and cancer outcomes, providing an overall perspective that aids in evaluating patients' risk levels or trends within specific populations. Future research is warranted to validate these findings and develop clinically applicable models to select high-risk patients for specific cancers within particular populations.

Below, we present our responses and revisions.

Further, there are some points which should be clearly addressed.

Point 1

1. The authors used three methods of DNAm-derived LCR estimation. Are they gold standard? If not, they should describe why they selected them.

Response

We selected the deconvolution algorithms by focusing on those that are publicly available and specifically developed for whole blood samples (10-12). Algorithms that are not publicly available (13), not applicable to whole blood (e.g., developed for cord blood cell types) (14-17), or only compatible with the DNAm 450K array (18), which would limit their applicability to our EPIC-based datasets (subsets I and II), were excluded. The choice of Houseman's, Salas', and LOLIPOP algorithms ensures applicability to our study population and compatibility with the methylation platform used, thereby supporting robust and reliable estimation of DNAm-derived leukocyte composition ratios.

We have added this explanation to the Methods section (Page 5, Lines 1–2).

Point 2

2. In the Discussion section, the authors mentioned the absence of cytologic LCR measurements. However, they should validate DNAm-derived LCR compositions using different experimental methods and evaluate differences among them.

Response

We appreciate the reviewer’s valuable comment. Unfortunately, direct leukocyte counts were not collected as part of the ESTHER study design, and therefore, cytologic LCRs could not be calculated in this cohort. However, in our previous work within a subset of the KAROLA (German: Langzeiterfolge der kardi ologischen Anschlussheilbehandlung) study, where both leukocyte counts and DNAm data were available, we found that the correlations between measured cell counts and DNAm-estimated cell proportions were high (12). These findings support the validity of DNAm-derived LCRs, though we acknowledge the importance of further cross-validation in future studies.

Point 3

3. Why DNAm-derived LCRs more strongly associate with lung cancer mortality compared with others?

Response

DNAm-derived LCRs reflect chronic systemic inflammation, which plays a critical role in cancer initiation, progression, and survival (6, 19). Although stronger associations were observed for lung cancer mortality in this study, similar trends were also noted for several other cancer types (20-23). However, due to the relatively small number of specific-cancer deaths beyond lung cancer in the ESTHER study (Response Table 2), statistical power is limited, which may lead to imprecise estimates and underestimation of true associations in these subgroups.

Response Table 2. Death event of cancers in ESTHER study and its subset

Death event	Overall	Subset 1	Subset 2				Subset 3			Subset 4
			Cancer Incidence case	Random death case	Random sub-cohort	Total	Death Case	Random sub-cohort	Total	
Total number	4529	1034	260	542	739	1541	406	548	954	1000
All-cause	2081	273	138	542	193	873	406	176	582	353
All-cancer	781	79	116	173	60	349	170	66	236	117
Lung cancer	233	22	60	44	18	122	44	18	62	27
Colorectal cancer	82	9	14	13	7	34	16	8	24	15
Pancreatic cancer	73	4	29	14	6	49	10	3	13	7
Breast cancer	51	2	3	15	5	23	11	7	18	8
Prostate cancer	40	6	1	12	5	18	7	2	9	7
Liver cancer	15	2	0	2	1	3	5	3	8	2
Ovarian cancer	26	3	0	7	2	9	7	2	9	5
Esophagus cancer	23	2	3	8	2	13	5	1	6	2
Head-neck cancer	17	5	0	4	1	5	5	1	6	1
Gastric cancer	12	2	0	1	0	1	5	1	6	3

In addition, we further highlight the relevance of DNAm-derived LCRs and chronic inflammation in lung cancer mortality and the potential underlying mechanisms. This discussion has been added to the revised manuscript.

Discussion section, Page 9, line 38-48:

Notably, strong associations between cytological NLR and lung cancer-specific mortality have also been observed, even in individuals without a prior lung cancer diagnosis (24). In clinical settings, elevated NLR, LMR, and other inflammation-related markers are widely used as prognostic indicators for lung cancer, particularly among patients with stage IV non-small cell lung cancer (25). From a mechanistic perspective, chronic exposure to airborne pathogens or toxic agents can trigger overproduction of reactive oxygen/nitrogen species (ROS/RNS), leading to sustained inflammation and lung tissue injury (26). Consistent with this, chronic inflammatory states—such as smoking, chronic obstructive pulmonary disease (COPD), and bronchitis—are well-established risk factors for lung cancer and may contribute to both aberrant methylation and cancer progression through shared molecular mechanisms (27-29).

Point 4

4. For Data availability, the authors could open the processed data (e.g. calculated LCRs).

Response

We appreciate the reviewer's suggestion. However, in accordance with the data protection policies of the German Cancer Research Center (DKFZ), we are unable to make the processed data publicly available. Nevertheless, the data can be made available upon reasonable request from the corresponding author, subject to institutional and ethical approvals.

References

1. Huan T, Nguyen S, Colicino E, Ochoa-Rosales C, Hill WD, Brody JA, et al. Integrative analysis of clinical and epigenetic biomarkers of mortality. *Aging Cell*. 2022;21(6):e13608.
2. Zhao Z, Bhardwaj M, Fan Z, Li X, Schrotz-King P, Brenner H. Smoking-independent DNA methylation markers for lung cancer risk: External validation in a large population-based cohort study. *Cancer Sci*. 2025;116(3):775-82.
3. Bhardwaj M, Schöttker B, Holleczer B, Brenner H. Enhanced selection of people for lung cancer screening using AHRH (cg05575921) or F2RL3 (cg03636183) methylation as biological markers of smoking exposure. *Cancer Commun (Lond)*. 2023;43(8):956-9.
4. Yu H, Raut JR, Schöttker B, Holleczer B, Zhang Y, Brenner H. Individual and joint contributions of genetic and methylation risk scores for enhancing lung cancer risk stratification: data from a population-based cohort in Germany. *Clin Epigenetics*. 2020;12(1):89.
5. Yamamoto T, Kawada K, Obama K. Inflammation-Related Biomarkers for the Prediction of Prognosis in Colorectal Cancer Patients. *Int J Mol Sci*. 2021;22(15).
6. Pellegrino R, Paganelli R, Di Iorio A, Bandinelli S, Moretti A, Iolascon G, et al. Temporal trends, sex differences, and age-related disease influence in Neutrophil, Lymphocyte count and Neutrophil to Lymphocyte-ratio: results from InCHIANTI follow-up study. *Immun Ageing*. 2023;20(1):46.
7. Templeton AJ, McNamara MG, Seruga B, Vera-Badillo FE, Aneja P, Ocana A, et al. Prognostic role of neutrophil-to-lymphocyte ratio in solid tumors: a systematic review and meta-analysis. *J Natl Cancer Inst*. 2014;106(6):dju124.

8. Nishijima TF, Muss HB, Shachar SS, Tamura K, Takamatsu Y. Prognostic value of lymphocyte-to-monocyte ratio in patients with solid tumors: A systematic review and meta-analysis. *Cancer Treat Rev.* 2015;41(10):971-8.
9. Chan SWS, Smith E, Aggarwal R, Balaratnam K, Chen R, Hueniken K, et al. Systemic Inflammatory Markers of Survival in Epidermal Growth Factor-Mutated Non-Small-Cell Lung Cancer: Single-Institution Analysis, Systematic Review, and Meta-analysis. *Clin Lung Cancer.* 2021;22(5):390-407.
10. Houseman EA, Accomando WP, Koestler DC, Christensen BC, Marsit CJ, Nelson HH, et al. DNA methylation arrays as surrogate measures of cell mixture distribution. *BMC Bioinformatics.* 2012;13:86.
11. Salas LA, Koestler DC, Butler RA, Hansen HM, Wiencke JK, Kelsey KT, et al. An optimized library for reference-based deconvolution of whole-blood biospecimens assayed using the Illumina HumanMethylationEPIC BeadArray. *Genome Biology.* 2018;19(1):64.
12. Heiss JA, Breitling LP, Lehne B, Kooner JS, Chambers JC, Brenner H. Training a model for estimating leukocyte composition using whole-blood DNA methylation and cell counts as reference. *Epigenomics.* 2017;9(1):13-20.
13. Salas LA, Zhang Z, Koestler DC, Butler RA, Hansen HM, Molinaro AM, et al. Enhanced cell deconvolution of peripheral blood using DNA methylation for high-resolution immune profiling. *Nature Communications.* 2022;13(1):761.
14. Bakulski KM, Feinberg JI, Andrews SV, Yang J, Brown S, S LM, et al. DNA methylation of cord blood cell types: Applications for mixed cell birth studies. *Epigenetics.* 2016;11(5):354-62.
15. de Goede OM, Razzaghian HR, Price EM, Jones MJ, Kobor MS, Robinson WP, et al. Nucleated red blood cells impact DNA methylation and expression analyses of cord blood hematopoietic cells. *Clinical Epigenetics.* 2015;7(1):95.
16. Gervin K, Salas LA, Bakulski KM, van Zelm MC, Koestler DC, Wiencke JK, et al. Systematic evaluation and validation of reference and library selection methods for deconvolution of cord blood DNA methylation data. *Clinical Epigenetics.* 2019;11(1):125.
17. Gervin K, Page CM, Aass HC, Jansen MA, Fjeldstad HE, Andreassen BK, et al. Cell type specific DNA methylation in cord blood: A 450K-reference data set and cell count-based validation of estimated cell type composition. *Epigenetics.* 2016;11(9):690-8.
18. Reinius LE, Acevedo N, Joerink M, Pershagen G, Dahlén SE, Greco D, et al. Differential DNA methylation in purified human blood cells: implications for cell lineage and studies on disease susceptibility. *PLoS One.* 2012;7(7):e41361.
19. Greten FR, Grivennikov SI. Inflammation and Cancer: Triggers, Mechanisms, and Consequences. *Immunity.* 2019;51(1):27-41.
20. Wiencke JK, Koestler DC, Salas LA, Wiemels JL, Roy RP, Hansen HM, et al. Immunomethylomic approach to explore the blood neutrophil lymphocyte ratio (NLR) in glioma survival. *Clin Epigenetics.* 2017;9:10.
21. Koestler DC, Usset J, Christensen BC, Marsit CJ, Karagas MR, Kelsey KT, et al. DNA Methylation-Derived Neutrophil-to-Lymphocyte Ratio: An Epigenetic Tool to Explore Cancer Inflammation and Outcomes. *Cancer Epidemiol Biomarkers Prev.* 2017;26(3):328-38.
22. Zhao N, Ruan M, Koestler DC, Lu J, Salas LA, Kelsey KT, et al. Methylation-derived inflammatory measures and lung cancer risk and survival. *Clin Epigenetics.* 2021;13(1):222.
23. Arroyo VM, Lupo PJ, Scheurer ME, Rednam SP, Murray J, Okcu MF, et al. Pilot study of DNA methylation-derived neutrophil-to-lymphocyte ratio and survival in pediatric medulloblastoma. *Cancer Epidemiol.* 2019;59:71-4.
24. Kang J, Chang Y, Ahn J, Oh S, Koo DH, Lee YG, et al. Neutrophil-to-lymphocyte ratio and risk of lung cancer mortality in a low-risk population: A cohort study. *Int J Cancer.* 2019;145(12):3267-75.
25. Mandalija H, Jones M, Oldmeadow C, Nordman, II. Prognostic biomarkers in stage IV non-small cell lung cancer (NSCLC): neutrophil to lymphocyte ratio (NLR), lymphocyte to monocyte ratio (LMR), platelet to lymphocyte ratio (PLR) and advanced lung cancer inflammation index (ALI). *Transl Lung Cancer Res.* 2019;8(6):886-94.
26. Azad N, Rojanasakul Y, Vallyathan V. Inflammation and lung cancer: roles of reactive oxygen/nitrogen species. *J Toxicol Environ Health B Crit Rev.* 2008;11(1):1-15.
27. Lee G, Walser TC, Dubinett SM. Chronic inflammation, chronic obstructive pulmonary disease, and lung cancer. *Curr Opin Pulm Med.* 2009;15(4):303-7.
28. Ahmad S, Manzoor S, Siddiqui S, Mariappan N, Zafar I, Ahmad A, et al. Epigenetic underpinnings of inflammation: Connecting the dots between pulmonary diseases, lung cancer and COVID-19. *Semin Cancer Biol.* 2022;83:384-98.

29. Brody JS, Spira A. State of the art. Chronic obstructive pulmonary disease, inflammation, and lung cancer. *Proc Am Thorac Soc.* 2006;3(6):535-7.

Point-by-point Response to the reviewers' comments of

Association of Leukocyte Composition Ratios from Blood Methylation with Cancer Mortality Outcomes: A Prospective Cohort Study

Reviewer #1

The authors present an analysis of DNA-based cell typing results from methylation deconvolution to test associations with mortality in a large prospective cohort using multiple subsets of participants. This large study makes important contributions to assessing the potential utility of DNA-based cell typing with DNA methylation measures and takes advantage of participants with long-term follow up data in the ESTHER cohort. Data analyses are rigorous and include appropriate consideration of potential confounding in multivariable models. Authors also include sensitivity analyses to restrict based on follow up time and also show analyses stratified by biological sex. The findings provide key information about capabilities of DNA methylation deconvolution that offer additional leukocyte subtypes compared with traditional clinical cell enumeration methods. I have no major concerns. It appears that the Sala's algorithm may actually be Salas' ?

Response

We are grateful to the reviewer for recognising the merits in our manuscript. We acknowledge the mistake use of Sala's and have corrected all the Sala's to Salas' accordingly.

Reviewer #2

The manuscript investigates the prognostic value of DNAm-derived lymphocyte-to-cell ratios (LCRs) in predicting all-cause, cancer-specific, and lung cancer-specific mortality. The study is based on a large prospective cohort (ESTHER study, n=3,886) with 17 years of follow-up, providing longitudinal insights into the association between DNAm-derived inflammatory markers and mortality risks. While the study presents important findings, several points could be clarified or expanded to enhance its rigor and potential clinical relevance:

Response

We express our appreciation to the reviewer for recognizing the strengths of our manuscript. Below, we present our point-to-point responses and revisions.

Point 1

1. Although DNAm markers may precede disease progression, they could also be influenced by pre-existing health conditions. Could the authors clarify how they address this potential reverse causation issue? Have they considered additional health conditions that might contribute to altered DNAm patterns?

Response

Thank you for this important comment. In our previous integrative analyses of clinical and epigenetic biomarkers of mortality—including cardiovascular and cancer-specific deaths—we found that age, sex, BMI, smoking, and alcohol consumption are well-established pre-existing health conditions that strongly predict cancer mortality (1). Furthermore, smoking emerged as one of the most influential factors associated with DNA methylation, particularly in the context of lung cancer (2-4). In the current analysis, we adjusted for comprehensive covariates including age, sex, BMI, smoking and alcohol consumption in the Cox regression models. Given that DNAm-derived LCRs may reflect systemic inflammatory status, we further adjusted for chronic inflammation-related conditions such as diabetes, hypertension, and a history of cardiovascular events. In addition, blood samples were obtained at the time of the interview, and these collections were not conducted during acute infections. These adjustments help mitigate the potential influence of pre-existing health conditions and reduce the likelihood of reverse causation.

Point 2

2. The authors have adjusted for smoking pack-years, but could other lifestyle and comorbidity factors also influence mortality risk? Additional sensitivity analyses considering chronic diseases (e.g., kidney disease, chronic inflammation, autoimmune conditions, or medication use) would help assess whether these factors modify the associations between DNAm-derived LCRs and mortality outcomes.

Response

Thank you for this insightful comment. In our primary multivariable models, we adjusted for a range of sociodemographic and lifestyle-related factors, as well as chronic inflammation-related conditions, including diabetes, hypertension, and history of cardiovascular disease.

To address the reviewer’s suggestion more directly, we conducted additional sensitivity analyses by further adjusting—separately—for renal failure, asthma, chronic obstructive pulmonary disease (COPD), rheumatoid arthritis, neurodermatitis, and NSAID (Table 1) use in the multivariable Cox regression models examining associations between DNAm-derived LCRs and 17-year all-cause, cancer-specific, and lung cancer–specific mortality (see Supplementary Data 1–3).

However, due to a high rate of missing data for renal failure, COPD, and rheumatoid arthritis, these models are likely subject to considerable bias. Additionally, for some models, the small number of corresponding death events led to non-convergence of the Cox regression. For the sensitivity analyses additionally adjusted for asthma, neurodermatitis, and NSAID use, we observed similar results compared to the results from the comprehensively adjusted models. These findings support the robustness and stability of our main results.

Table 1. Characteristics of the study population for additional variates in sensitivity analysis

Characteristics	Overall (N = 3886)	Subset I (N=935)	Subset II (N=1117)	Subset III (N= 857)	Subset IV (N=977)
Prevalent renal failure (N, %) a	30 (2.1)	3 (0.7)	13 (3.2)	5 (2.1)	9 (2.3)
Prevalent asthma (N, %) b	233 (6.2)	48 (5.3)	70 (6.4)	40 (4.9)	75 (7.9)
Prevalent COPD (N, %) c	16 (1.1)	4 (0.9)	5 (1.3)	3 (1.2)	4 (1.0)
Prevalent rheumatoid arthritis (N, %) d	9 (0.7)	2 (0.5)	2 (0.6)	2 (1.0)	3 (0.9)
Prevalent neurodermatitis (N, %) e	142 (3.9)	33 (3.7)	46 (4.4)	30 (3.7)	33 (3.5)
Prevalent NSAIDs (N, %) f					
Prescribed Aspirin only	471 (12.1)	109 (11.7)	137 (12.3)	111 (13.0)	114 (11.7)
Prescribed other NSAID	137 (3.5)	24 (2.6)	43 (3.9)	27 (3.2)	43 (4.4)
Daily use of non-prescribed pain killer	204 (5.3)	47 (5.0)	69 (6.2)	42 (4.9)	46 (4.7)

^a Data missing for 513, 714, 624, 580, and 2431 participants in subset I, II, III, IV, and across all study populations in ESTHER.

^b Data missing for 23, 30, 38, 22, and 113 in subset I, II, III, IV, and across all study populations in ESTHER.

^c Data missing for 510, 719, 616, 586, and 2431 in subset I, II, III, IV, and across all study populations in ESTHER.

^d Data missing for 567, 761, 654, 633, and 2615 in subset I, II, III, IV, and across all study populations in ESTHER.

^e Data missing for 47, 73, 56, 43 and 219 in subset I, II, III, IV, and across all study populations in ESTHER.

^f Data missing for 1 and 1 in subset II and across all study populations in ESTHER.

Point 3

3. The findings suggest that DNAm-derived leukocyte composition ratios (LCRs) are strongly associated with all-cause, cancer-specific, and lung cancer-specific mortality. Given these significant associations. Could the authors discuss whether these DNAm-derived markers could serve as potential tools for clinical risk stratification? Are there thresholds or cutoffs for these LCRs that could be used for stratifying high-risk patients in a clinical setting?

Response

Given the known variability of LCR in optimal cutoffs across populations, even widely studied ratios such as NLR and LMR have shown a broad range of thresholds (e.g., between 2 and 5) (5). In addition, the ESTHER study includes four distinct subsets, each with differing baseline characteristics, which further complicates the definition of a universal cutoff. Therefore, we adopted a quartile-based approach to explore dose-response relationships between DNAm-derived LCRs and cancer outcomes. This strategy allows for consistent interpretation without relying on potentially unstable absolute thresholds. Future research will be essential to validate these markers for clinical risk stratification and to define standardized cutoffs.

Point 4

4. In the methods and results, authors mentioned 44 DNAm-derived LCRs, I suppose that authors should have their reasons to select those 44 LCRs? Could authors explain the reason and add it to in the manuscript to make it more clear for the readers?

Response

Thank you for this important question. The 44 DNAm-derived leukocyte composition ratios (LCRs) were selected based on three deconvolution algorithms that estimate proportions of granulocytes, monocytes, lymphocytes, and lymphocyte subtypes. These LCRs include key inflammation-related ratios such as neutrophil-to-lymphocyte ratio (NLR), lymphocyte-to-monocyte ratio (LMR), sum of neutrophils and monocytes to lymphocytes ratio (sNMLR), and product of neutrophils and monocytes to lymphocytes ratio (pNMLR), which have been previously associated with elevated mortality in cancer patients (6-9). This rationale has now been added to the Methods section (Page 5, Lines 19-21) for clarity.

Point 5

5. In the methods part (lines:134-136), "We trimmed the leukocyte composition values by setting the outliers to the upper bound", then how're the outliers?" Were they excluded or replaced?

Response

The outliers were not excluded but replaced. Specifically, values exceeding the upper bound (defined as the third quartile + $1.5 \times$ interquartile range [IQR]) were set to the upper bound, and values below the lower bound (first quartile - $1.5 \times$ IQR) were set

to the lower bound. We have now clarified this procedure in the Methods section (Page 5, Lines 23-24).

Reviewer #2 (2nd Round)

The manuscript presents an insightful and rigorous investigation into the association between DNAm-derived leukocyte composition ratios (LCRs) and multiple mortality outcomes over a 17-year follow-up within a large prospective cohort. The use of epigenome-wide data to quantify systemic inflammation and its prognostic value is both novel and compelling. The major strengths include the long follow-up duration, comprehensive covariate adjustment, and stratified analyses across demographic subgroups.

The study's findings—especially the robust associations of NLR, LMR, and neutrophil-to-B-cell ratios with all-cause, cancer-specific, and lung cancer-specific mortality—are convincing and well-supported by sensitivity analyses. The consistency of hazard ratio estimates across multiple follow-up intervals enhances the credibility of the results.

The authors have addressed potential confounding and reverse causation appropriately. Their adjustments for chronic diseases and inflammation-related conditions, as well as the use of sensitivity analyses to account for comorbidities such as renal failure, asthma, and medication use, further reinforce the robustness of the findings. While some models were limited by missing data and low event counts, the consistency of results supports the main conclusions.

The decision to use quartile-based LCRs for dose-response analyses instead of predefined thresholds is justified and reflects a cautious approach given the variability in cutoffs across populations. This method enhances interpretability and facilitates future validation work.

Clarifications provided about the selection of 44 LCRs from three deconvolution algorithms and the trimming method for outlier handling have improved the transparency and reproducibility of the methods.

Overall, the study makes a valuable contribution to the field of epigenetic epidemiology and cancer prognostication. With minor revisions as incorporated, the manuscript is suitable for publication.

Reviewer #3

In this paper, Ziwen Fan et al. conducted calculation of Leukocyte Composition Ratios (LCR) from DNA methylation data of blood samples and evaluated associations between DNAm-derived LCR and cancer mortality. The authors found that DNAm-derived LCRs were associated with all-cause mortality. Several kinds of ratios were also correlated with mortality. The data with long-term follow-up is very valuable. However, the reviewer thought that the authors at least need to show models to select high-risk patients for enriching clinical applications based on DNAm-derived LCRs because biological relevance was not enough in the findings of this paper.

Response

Thank you for your thoughtful feedback on our study.

We acknowledge the importance of identifying high-risk patients for clinical applications. However, due to the known variability of optimal LCR cutoffs across populations and the heterogeneity among the four distinct subgroups in the ESTHER study, establishing a model with a universal threshold for risk stratification is currently challenging. Therefore, we employed a quartile-based approach to capture dose-response relationships between DNAm-derived LCRs and cancer outcomes, providing an overall perspective that aids in evaluating patients' risk levels or trends within specific populations. Future research is warranted to validate these findings and develop clinically applicable models to select high-risk patients for specific cancers within particular populations.

Below, we present our responses and revisions.

Further, there are some points which should be clearly addressed.

Point 1

1. The authors used three methods of DNAm-derived LCR estimation. Are they gold standard? If not, they should describe why they selected them.

Response

We selected the deconvolution algorithms by focusing on those that are publicly available and specifically developed for whole blood samples (10-12). Algorithms that are not publicly available (13), not applicable to whole blood (e.g., developed for cord blood cell types) (14-17), or only compatible with the DNAm 450K array (18), which would limit their applicability to our EPIC-based datasets (subsets I and II), were excluded. The choice of Houseman's, Salas', and LOLIPOP algorithms ensures applicability to our study population and compatibility with the methylation platform used, thereby supporting robust and reliable estimation of DNAm-derived leukocyte composition ratios.

We have added this explanation to the Methods section (Page 5, Lines 1–2).

Point 2

2. In the Discussion section, the authors mentioned the absence of cytologic LCR measurements. However, they should validate DNAm-derived LCR compositions using different experimental methods and evaluate differences among them.

Response

We appreciate the reviewer's valuable comment. Unfortunately, direct leukocyte counts were not collected as part of the ESTHER study design, and therefore, cytologic LCRs could not be calculated in this cohort. However, in our previous work within a subset of the KAROLA (German: Langzeiterfolge der kardi ologischen Anschlussheilbehandlung) study, where both leukocyte counts and DNAm data were available, we found that the correlations between measured cell counts and DNAm-estimated cell proportions were high (12). These findings support the validity of DNAm-derived LCRs, though we acknowledge the importance of further cross-validation in future studies.

Point 3

3. Why DNAm-derived LCRs more strongly associate with lung cancer mortality compared with others?

Response

DNAm-derived LCRs reflect chronic systemic inflammation, which plays a critical role in cancer initiation, progression, and survival (6, 19). Although stronger associations were observed for lung cancer mortality in this study, similar trends were also noted for several other cancer types (20-23). However, due to the relatively small number of specific-cancer deaths beyond lung cancer in the ESTHER study, statistical power is limited, which may lead to imprecise estimates and underestimation of true associations in these subgroups.

In addition, we further highlight the relevance of DNAm-derived LCRs and chronic inflammation in lung cancer mortality and the potential underlying mechanisms. This discussion has been added to the revised manuscript.

Discussion section, Page 9, line 38-48:

Notably, strong associations between cytological NLR and lung cancer-specific mortality have also been observed, even in individuals without a prior lung cancer diagnosis (24). In clinical settings, elevated NLR, LMR, and other inflammation-related markers are widely used as prognostic indicators for lung cancer, particularly among patients with stage IV non-small cell lung cancer (25). From a mechanistic perspective, chronic exposure to airborne pathogens or toxic agents can trigger overproduction of reactive oxygen/nitrogen species (ROS/RNS), leading to sustained inflammation and lung tissue injury (26). Consistent with this, chronic inflammatory states—such as smoking, chronic obstructive pulmonary disease (COPD), and bronchitis—are well-established risk factors for lung cancer and may contribute to both aberrant methylation and cancer progression through shared molecular mechanisms (27-29).

Point 4

4. For Data availability, the authors could open the processed data (e.g. calculated LCRs).

Response

We appreciate the reviewer's suggestion. However, in accordance with the data protection policies of the German Cancer Research Center (DKFZ), we are unable to make the processed data publicly available. Nevertheless, the data can be made available upon reasonable request from the corresponding author, subject to institutional and ethical approvals.

Reviewer #3 (2nd Round)

The reviewer understood that it was challenging to establish a model. The authors' approach is more practical to capture the relationship between DNAm-derived LCRs and outcomes. They have addressed all of the concerns.

References

1. Huan T, Nguyen S, Colicino E, Ochoa-Rosales C, Hill WD, Brody JA, et al. Integrative analysis of clinical and epigenetic biomarkers of mortality. *Aging Cell*. 2022;21(6):e13608.
2. Zhao Z, Bhardwaj M, Fan Z, Li X, Schrotz-King P, Brenner H. Smoking-independent DNA methylation markers for lung cancer risk: External validation in a large population-based cohort study. *Cancer Sci*. 2025;116(3):775-82.
3. Bhardwaj M, Schöttker B, Holleczer B, Brenner H. Enhanced selection of people for lung cancer screening using AHRH (cg05575921) or F2RL3 (cg03636183) methylation as biological markers of smoking exposure. *Cancer Commun (Lond)*. 2023;43(8):956-9.
4. Yu H, Raut JR, Schöttker B, Holleczer B, Zhang Y, Brenner H. Individual and joint contributions of genetic and methylation risk scores for enhancing lung cancer risk stratification: data from a population-based cohort in Germany. *Clin Epigenetics*. 2020;12(1):89.
5. Yamamoto T, Kawada K, Obama K. Inflammation-Related Biomarkers for the Prediction of Prognosis in Colorectal Cancer Patients. *Int J Mol Sci*. 2021;22(15).
6. Pellegrino R, Paganelli R, Di Iorio A, Bandinelli S, Moretti A, Iolascon G, et al. Temporal trends, sex differences, and age-related disease influence in Neutrophil, Lymphocyte count and Neutrophil to Lymphocyte-ratio: results from InCHIANTI follow-up study. *Immun Ageing*. 2023;20(1):46.
7. Templeton AJ, McNamara MG, Seruga B, Vera-Badillo FE, Aneja P, Ocana A, et al. Prognostic role of neutrophil-to-lymphocyte ratio in solid tumors: a systematic review and meta-analysis. *J Natl Cancer Inst*. 2014;106(6):dju124.
8. Nishijima TF, Muss HB, Shachar SS, Tamura K, Takamatsu Y. Prognostic value of lymphocyte-to-monocyte ratio in patients with solid tumors: A systematic review and meta-analysis. *Cancer Treat Rev*. 2015;41(10):971-8.
9. Chan SWS, Smith E, Aggarwal R, Balaratnam K, Chen R, Hueniken K, et al. Systemic Inflammatory Markers of Survival in Epidermal Growth Factor-Mutated Non-Small-Cell Lung Cancer: Single-Institution Analysis, Systematic Review, and Meta-analysis. *Clin Lung Cancer*. 2021;22(5):390-407.
10. Houseman EA, Accomando WP, Koestler DC, Christensen BC, Marsit CJ, Nelson HH, et al. DNA methylation arrays as surrogate measures of cell mixture distribution. *BMC Bioinformatics*. 2012;13:86.
11. Salas LA, Koestler DC, Butler RA, Hansen HM, Wiencke JK, Kelsey KT, et al. An optimized library for reference-based deconvolution of whole-blood biospecimens assayed using the Illumina HumanMethylationEPIC BeadArray. *Genome Biology*. 2018;19(1):64.
12. Heiss JA, Breitling LP, Lehne B, Kooner JS, Chambers JC, Brenner H. Training a model for estimating leukocyte composition using whole-blood DNA methylation and cell counts as reference. *Epigenomics*. 2017;9(1):13-20.

13. Salas LA, Zhang Z, Koestler DC, Butler RA, Hansen HM, Molinaro AM, et al. Enhanced cell deconvolution of peripheral blood using DNA methylation for high-resolution immune profiling. *Nature Communications*. 2022;13(1):761.
14. Bakulski KM, Feinberg JI, Andrews SV, Yang J, Brown S, S LM, et al. DNA methylation of cord blood cell types: Applications for mixed cell birth studies. *Epigenetics*. 2016;11(5):354-62.
15. de Goede OM, Razzaghian HR, Price EM, Jones MJ, Kobor MS, Robinson WP, et al. Nucleated red blood cells impact DNA methylation and expression analyses of cord blood hematopoietic cells. *Clinical Epigenetics*. 2015;7(1):95.
16. Gervin K, Salas LA, Bakulski KM, van Zelm MC, Koestler DC, Wiencke JK, et al. Systematic evaluation and validation of reference and library selection methods for deconvolution of cord blood DNA methylation data. *Clinical Epigenetics*. 2019;11(1):125.
17. Gervin K, Page CM, Aass HC, Jansen MA, Fjeldstad HE, Andreassen BK, et al. Cell type specific DNA methylation in cord blood: A 450K-reference data set and cell count-based validation of estimated cell type composition. *Epigenetics*. 2016;11(9):690-8.
18. Reinius LE, Acevedo N, Joerink M, Pershagen G, Dahlén SE, Greco D, et al. Differential DNA methylation in purified human blood cells: implications for cell lineage and studies on disease susceptibility. *PLoS One*. 2012;7(7):e41361.
19. Gretten FR, Grivennikov SI. Inflammation and Cancer: Triggers, Mechanisms, and Consequences. *Immunity*. 2019;51(1):27-41.
20. Wiencke JK, Koestler DC, Salas LA, Wiemels JL, Roy RP, Hansen HM, et al. Immunomethylomic approach to explore the blood neutrophil lymphocyte ratio (NLR) in glioma survival. *Clin Epigenetics*. 2017;9:10.
21. Koestler DC, Usset J, Christensen BC, Marsit CJ, Karagas MR, Kelsey KT, et al. DNA Methylation-Derived Neutrophil-to-Lymphocyte Ratio: An Epigenetic Tool to Explore Cancer Inflammation and Outcomes. *Cancer Epidemiol Biomarkers Prev*. 2017;26(3):328-38.
22. Zhao N, Ruan M, Koestler DC, Lu J, Salas LA, Kelsey KT, et al. Methylation-derived inflammatory measures and lung cancer risk and survival. *Clin Epigenetics*. 2021;13(1):222.
23. Arroyo VM, Lupo PJ, Scheurer ME, Rednam SP, Murray J, Okcu MF, et al. Pilot study of DNA methylation-derived neutrophil-to-lymphocyte ratio and survival in pediatric medulloblastoma. *Cancer Epidemiol*. 2019;59:71-4.
24. Kang J, Chang Y, Ahn J, Oh S, Koo DH, Lee YG, et al. Neutrophil-to-lymphocyte ratio and risk of lung cancer mortality in a low-risk population: A cohort study. *Int J Cancer*. 2019;145(12):3267-75.
25. Mandaliya H, Jones M, Oldmeadow C, Nordman, II. Prognostic biomarkers in stage IV non-small cell lung cancer (NSCLC): neutrophil to lymphocyte ratio (NLR), lymphocyte to monocyte ratio (LMR), platelet to lymphocyte ratio (PLR) and advanced lung cancer inflammation index (ALI). *Transl Lung Cancer Res*. 2019;8(6):886-94.
26. Azad N, Rojanasakul Y, Vallyathan V. Inflammation and lung cancer: roles of reactive oxygen/nitrogen species. *J Toxicol Environ Health B Crit Rev*. 2008;11(1):1-15.
27. Lee G, Walser TC, Dubinett SM. Chronic inflammation, chronic obstructive pulmonary disease, and lung cancer. *Curr Opin Pulm Med*. 2009;15(4):303-7.
28. Ahmad S, Manzoor S, Siddiqui S, Mariappan N, Zafar I, Ahmad A, et al. Epigenetic underpinnings of inflammation: Connecting the dots between pulmonary diseases, lung cancer and COVID-19. *Semin Cancer Biol*. 2022;83:384-98.
29. Brody JS, Spira A. State of the art. Chronic obstructive pulmonary disease, inflammation, and lung cancer. *Proc Am Thorac Soc*. 2006;3(6):535-7.